# A Review of Delayed Delivery Models and the Analysis Method in Mice

**DOI:** 10.3390/jdb10020020

**Published:** 2022-05-20

**Authors:** Hiroshi Yomogita, Naoyuki Miyasaka, Masami Kanai-Azuma

**Affiliations:** 1Department of Perinatal and Women’s Medicine, Tokyo Medical and Dental University, Tokyo 113-8510, Japan; yomocrm@tmd.ac.jp (H.Y.); n.miyasaka.gyne@tmd.ac.jp (N.M.); 2Center for Experimental Animals, Tokyo Medical and Dental University, Tokyo 113-8510, Japan

**Keywords:** prolonged delivery, delayed delivery, progesterone, luteolysis

## Abstract

In humans, the incidence of post-term delivery is 1–10%. Post-term delivery significantly increases the risk of cesarean section or neonatal intensive care unit (NICU) admission. Despite these serious challenges, the cause of prolonged delivery remains unclear. Several common factors of delayed parturition between mice and humans will help elucidate the mechanisms of pregnancy and labor. At present, gene modification techniques are rapidly developing; however, there are limited reviews available describing the mouse phenotype analysis as a human model for post-term delivery. We classified the delayed-labor mice into nine types according to their causes. In mice, progesterone (P₄) maintains pregnancy, and the most common cause of delayed labor is luteolysis failure. Other contributing factors include humoral molecules in the fetus/placenta, uterine contractile dysfunction, poor cervical ripening, and delayed implantation. The etiology of delayed parturition is overexpression of the pregnancy maintenance mechanism or suppression of the labor induction mechanism. Here, we describe how to investigated their causes using mouse genetic analysis. In addition, we generated a list to identify the causes. Our review will help understand the findings obtained using the mouse model, providing a foundation for conducting more systematic research on delayed delivery.

## 1. Introduction

Placental–fetal–maternal close signal interactions are essential for pregnancy and labor. Imbalances between these factors cause abnormalities during the gestation period. Recently, several mice with delayed labor have been reported due to abnormal development of the fetus and placenta. In humans, the incidence of post-term delivery is 1–10% [1]. Post-term delivery significantly increases the risk of cesarean section or neonatal intensive care unit (NICU) admission. Despite these serious challenges, the cause of prolonged delivery remains unclear. Primates have similar-evolved villous placentas. In many studies, mouse pregnancy has been used as a model of human pregnancy. At present, gene modification techniques are rapidly developing; however, there are limited reviews to describe mouse phenotype analysis as a human model for post-term delivery. To elucidate, the effects of fetal and placental developmental abnormalities on delayed labor, we reviewed genetically modified mice that cause delayed labor. We reviewed the causes of delayed births and investigated them using mouse genetic analysis.

## 2. Normal Progression of Pregnancy and Labor Onset in Mice

Parturition is divided into two phases, pregnancy and labor, which correlate with changes in the cervix and myometrium (Figure 1). The cervix, a cylindrical tissue, is the lower part of the uterus and is used as the birth canal. During pregnancy, the cervix is firm and closed. It protects the fetus from microorganisms and acts as a barrier between intrauterine and extrauterine conditions. The cervix gradually softens as pregnancy progresses. The uterine myometrium is quiescent and does not respond to natural stimuli. In the labor phase, the cervix is soft and elastic, which is referred to as ‘cervical ripening’. Cervical ripening is caused by the decomposition of collagen and glycosaminoglycans (the main material of the cervix). The uterine myometrium contracts collaboratively and provides sufficient pressure to eliminate the fetus and placenta. Cervical ripening and uterine contractions are essential delivery processes, hence if either is missing, labor does not progress. The pregnancy-maintenance mechanism suppresses cervical ripening and uterine contractions. In the labor phase, the pregnancy-maintenance mechanism weakens, while the labor-induction mechanism is strengthened. Therefore, cervical ripening and uterine contractions occur. To deliver at term, the two mechanisms must switch at the right time. Delayed delivery occurs when the pregnancy-maintenance mechanism is excessively long or when the labor-induction mechanism is weak. A proper balance between the two systems is essential for normal pregnancy and delivery.

## 3. Factors of Pregnancy Maintenance and Labor Induction

In rodents, progesterone (P₄) has the strongest pregnancy-maintenance effect. P₄ is secreted by the corpus luteum in pregnant mice, and suppresses labor-induction substances, such as prostaglandins (PGs) and oxytocin receptor (OTR). At 18.5 days post coitum (dpc), serum P₄ declines rapidly because of luteolysis and P₄ withdrawal increases the levels of the labor-inducing substances (Figure 2).

Other substances that maintain pregnancy include relaxin and placental lactogen (PL), a part of the PRL family, which can maintain the corpus luteum. Human placental lactogen (hPL) does not maintain pregnancy. In contrast, mouse placental lactogen 1 (mPL1) and mouse placental lactogen 2 (mPL2) prevent luteolysis.

Labor-inducing substances include oxytocin, PGE₂ (prostaglandin E2), PGF2α (prostaglandin F2α), their receptors, and connexin (Cx). These are collectively referred to as contractile-associated proteins (CAPS).

Oxytocin, a hormone secreted by the pituitary gland, has a uterine contraction effect and is used as a labor-inducing agent in humans; however, *oxytocin* knockout (*OT* KO) mice can normally deliver [2]. Oxytocin is not essential for delivery. In oxytocin-administration experiments in *OT* KO mice, low concentrations of oxytocin maintained the corpus luteum and caused delayed labor and high concentrations of oxytocin enhanced uterine contractions and caused preterm delivery [3]. It is, therefore, difficult to elucidate the mechanism of labor since these molecules show different functions depending on the site and timing.

PG is a bioactive lipid produced from free arachidonic acid. PG is produced by phospholipase A (PLA) 2 and Cox; Cox1 and Cox2 are rate-limiting enzymes. There are various types of PG, such as D2, E2, F2α, and GI2, all of which have a different function. PGF2α and PGE₂ are used as labor-inducing agents in humans. 

PGF₂α also has a luteolytic effect in mice. PGE₂ has a cervical-ripening effect and causes weak uterine contractions. However, PGE₂ may also have an indirect luteotropic effect [4]. Since PG has a short half-life and acts locally around the secretory site, it is characterized by different functions depending on the site of action.

Cx is a protein that forms gap junctions, which are cell–cell conjugations. Gap junctions play a role in the exchange of calcium ions between cells. When Cx43 is abundantly expressed in the myometrium, gap junctions also increase. Gap junctions are important for cooperative uterine contractions.

## 4. Classification by Cause of Labor Delay

We searched the literature published between 1970 and 2022 and classified 26 mice according to the cause of delayed labor (Table 1).

### 4.1. Failure of Luteolysis

The most common cause of delayed delivery in genetically modified mice is the absence of P₄ withdrawal due to the failure of luteolysis. The decrease in serum P₄ can be confirmed by measuring approximately 18.5 dpc. Ovariectomy and administration of RU486 (a P₄ antagonist) are methods to determine whether P₄ maintains pregnancy [5,33]. Wild-type mice are delivered within 24 h using these methods [5,33].

There are two phases in the degeneration of the corpus luteum: functional and structural degeneration. Functional degeneration is associated with P₄ withdrawal. The enhanced action of 20α-hydroxysteroid dehydrogenase (20αHSD), which inactivates P₄, reduces serum P₄ levels. Meanwhile, in structural degeneration, apoptosis of pregnant luteal cells is enhanced, replacement of capillaries and connective tissue occurs, and the size and weight of the corpus luteum decreased rapidly. Structural degeneration can be confirmed by exenteration of the ovaries after 18.5 dpc and observing them with hematoxylin–eosin staining and/or tunnel staining.

#### 4.1.1. *Prostaglandin F Receptor* Knockout Mouse: *Fp* KO

One of the most famous genetically modified mice with delayed delivery is *Fp*-KO. In 1997, Sugimoto et al. produced mice to investigate the physiological function of PGF₂α [5]. *Fp* KO mice that lack the prostaglandin F receptor (FP) do not deliver and absorb their fetuses in utero [5]. The P₄ level of *Fp* KO mice did not decrease at 18.5 dpc [5]. *Fp* KO mice with both ovaries removed at 19.5 dpc delivered within 24 h [5]. Luteolysis failure is the cause of the delayed delivery. This report suggests that PGF2 leads to luteolysis in pregnant mice. 

In *Fp* KO at 19.5 dpc, ovariectomy leads to the expression of oxytocin, OTR, and Cox2 [6]. When Cox2 selective inhibitor was administered to these mice, labor did not occur [6]. Labor after luteolysis is caused by COX-2-derived PGs in the myometrium [6]. In wild-type mice, Cox2 selective inhibitor leads to prolonged delivery [34]. Conversely, in humans, Cox2 selective inhibitors restrict uterine contractions and have been investigated as a treatment to prevent preterm delivery [35]. 

#### 4.1.2. *Cox1*-Deficient Mouse: *Cox1* KO

The average gestation period of *Cox1* KO mice is 21.6 ± 0.2 days [7]. These mice showed no sharp P₄ decline and had a luteal regression disorder [7]. *Cox1* KO in late pregnancy results in a decreased amount of PGF2α in the uterus [7]. The administration of PGF2α to mice improved delayed delivery [7]. This research suggests that the cause of delayed parturition in mice is a failure of luteolysis due to a decrease in PGF₂α levels [7]. The uterine contractions of *Cox1* KO mice were the same as those of wild-type mice, but the cervix was immature [33]. The cause of delayed labor in mice may also be related to poor cervical ripening due to impaired PG production.

#### 4.1.3. *20α Hydroxysteroid Dehydrogenase*-Deficient Mice: 20α *Hsd* KO

20αHSD is an enzyme that inactivates P₄. *20α Hsd* KO caused delayed labor without P₄ withdrawal [8]. Administration of PGF2α did not induce labor in these mice [8]. 20αHSD is presumed to act downstream of PGF2α [8].

The expression of 20αHSD is controlled by various pathways (Figure 3). For example, PRL activates a transcription factor called Stat5b (signal transducer and activator of transcription 5 B) in the ovary [36]. Since Stat5b reduces the expression of 20αHSD, PRL has a luteotropic function. In mice, the lack of Stat5b is associated with early abortion [8].

#### 4.1.4. *Gαq^f/f^;Gα11^−/−^;Cre+* Mice: *Gq/11* cKO

Gαq/11 is a G protein that binds to receptors on the surface of activated cell membranes and transmits signals to cells. It has been speculated that FP activates the PLC pathway via Gαq/11 and activates 20αHSD in luteal cells [9]. *Gq/11* cKO labor was prolonged without P₄ withdrawal [9]. The mice showed decreased expression of *Akr1c18* (a gene encoding 20αHSD) in the ovary [9]. Therefore, FP was presumed to stimulate *Akr1c18* expression and increase 20αHSD via the Gq/11 phospholipase C pathway [9].

#### 4.1.5. *Mastermind-Like Domain-Containing 1* Knockout Mouse: *Mamld1* KO

Mamld1 is a gene that causes hypospadias in humans and its KO can delay parturition by approximately 50% [10]. In *Mamld1* KO, P₄ withdrawal does not exist, and Akr1c18 expression is decreased in the ovary [10]. Mamld1 suppresses *Akr1c18* by increasing Stat5 expression [10]. Therefore, these mice suffer from functional luteal regression disorders. Mamld1 is speculated to regress the corpus luteum by a different route to the FP pathway [10].

There are many regressive pathways in the corpus luteum, which are extremely complex. Further research is needed on the maintenance and disappearance of CL.

### 4.2. Abnormalities in P₄ Metabolism and Receptors 

The progesterone receptor (PR) has two different nuclear receptors, PRA and PRB, with different functions. PRA knockout mice are infertile, whereas PRB knockout mice suffer from mammary gland dysgenesis [37,38]. In humans, PRB suppresses Cx43, Cox2, OTR, and NFκβ pathways and maintains pregnancy [39]. However, PRA suppresses PRB and induces labor [39].

#### 4.2.1. *Kruppel-Like Factor 9* Knockout Mouse: *Klf* 9 KO

Kruppel-like factor 9 (KLF9) is a transcription factor for PRA and PRB (Figure 4). KLP9 deficiency in early pregnancy suppresses PR expression and reduces fertility [40]. The serum P₄ level of *Klf 9* KO mice were the same as that of wild-type mice, but the gestation of *Klf 9* KO mice was longer than that of wild-type mice [11]. In these mice, PRB expression is presumed to increase owing to the decreased expression of PRA [11]. KLF9 is expressed in humans and decreased in patients with delayed delivery [41]. This factor may be involved in the mechanisms of pregnancy and delivery in humans [41]. 

In humans, the corpus luteum disappears during the second trimester of pregnancy. Therefore, P₄ withdrawal was not observed. The switching of progesterone receptors from PRB to PRA in late pregnancy is presumed to trigger labor induction [39,42].

#### 4.2.2. *Cytochrome p450 Family 11a1*-Overexpression Mouse: *Cyp11a1* Tg

Cyp11a1 (P450scc) is an enzyme that produces P₄ from cholesterol. Cyp11a1 knockout mice are fatal because of their inability to produce steroids and their underdeveloped fetal adrenal glands [43]. Chien et al., created a *Cyp11a1* Tg that overexpresses Cyp11a1 [12]. The mice showed delayed maturation of the corpus luteum during early pregnancy, causing implantation abnormalities, placental dysplasia, and fetal development abnormalities [12]. The serum P₄ of *Cyp11a1* Tg mice was similar to that of wild-type mice in early pregnancy, but remained high in late pregnancy [12]. *Cyp11a1* Tg mice frequently show delayed parturition [12].

### 4.3. Fetal Factors

In autosomal dominant inheritance, crossing genetically modified females with male wild-type mice can determine whether the cause of delayed labor is the mother or the placenta/fetus. If the cause was the mother, the mice showed delayed delivery, while this does not occur if the cause was the placenta or the fetus. This can also be examined by transplanting KO embryos into wild-type mice [16].

#### 4.3.1. *Surfactant Protein A and D* Double-Deficient Mice: *Sp A/D* dKO

Lung surfactant protein (Sp) is a lipoprotein produced in the lungs that assists breathing. There are four types of SP, A to D. SP-A is the most abundant and structurally similar to SP-D. SP is produced in the fetal lungs and then flows into the amniotic fluid. In humans, fetal lung maturity can be estimated by examining the SP in the amniotic fluid. SPA acts as the trigger of labor, because administration of Sp-A to the amniotic fluid of mice results in preterm birth [13]. *Sp A/D* dKO showed delayed delivery for approximately 12 h in the second and subsequent pregnancies [13]. Mice at 18.5 dpc had decreased expression of CAPS in the uterine muscle and decreased secretion of inflammatory cytokines from macrophages in the amniotic fluid [13]. The hypothesis that the Sp of the fetal lung can be the starting point for labor induction is particularly important since the fetus itself is involved in determining the time of labor.

#### 4.3.2. *Steroid Receptor Coactivator 1 and 2* Double-Deficient Mice: *Src1/2* dKO

Steroid receptor coactivator 1 and 2 (Src1/2) (transcription factor of Sp-A) knockout mice showed prolonged delivery by 38 h [14]. These mice exhibited a reduction in Sp-A, PGF2α, Cx43, and OXTR, a restricted NF-κB pathway, and P₄ withdrawal did not occur [14]. Interestingly, the higher the proportion of *Src1/2* KO fetuses, the higher the serum P₄ concentration in late pregnancy [14]. 

In these mice, platelet-activating factor (PAF) in the fetal lung and amniotic fluid was also reduced [14]. PAF is involved in several physiological events, such as platelet aggregation, inflammation, and anaphylaxis, and is produced in the fetal lung; its concentration in the amniotic fluid increases during late pregnancy [44]. Therefore, PAF was presumed to play an important role in the onset of labor and preterm delivery. Administration of PAF to SRC-1/2 dKO increased NF-κB pathway activation and CAPS expression in the myometrium and improved delayed labor [14]. The results of this study suggest that PAF and SP-A may contribute to delivery.

### 4.4. Placental Factors

When the placenta is the cause, abnormal placental development often occurs, such as in Nrk KO and *Sirh7/Ldoc1* KO mice [15,16]. The lentiviral vector (LV) or tetraploid chimera can determine whether the cause is the placenta or fetus. When LV is administered to mouse blastocysts, the gene is transferred only to the placenta [45]. LV rescue delayed parturition due to placental factors. The tetraploid chimera utilizes the property that tetraploid embryonic cells contribute only to the placenta [46]. In chimeric mice established using wild-type diploid embryos and nuclear-transplanted cloned tetraploid embryos, delayed labor occurs when the cause is the placenta, but not when the cause is the fetus. In chimeras of wild-type tetraploid embryos and diploid embryos of nuclear-transplanted clones, the opposite result was observed.

#### 4.4.1. *Solute Carrier Organic Anion Transporter Family Member 2A1* Knockout Mouse: *Slco2a1* KO

*Slco2A1* encodes OATP2A1, an organic anion-transporting polypeptide. OATP2A1 is a PG transporter that takes up extracellular PGs. Mating between *Slco2A1* KO mice resulted in prolonged labor in approximately 50% of mice [4], and this did not occur after mating with wild-type male mice. This suggests that the cause is fetal or placental factors, hence P₄ withdrawal does not occur [4]. The level of mPL2 is elevated in the placenta [4]. mPL-2 is a member of the prolactin family and is known to have a luteotropic effect. In cultured placental cells of the sponge layer (one of the layers of the placenta), PGE₂ enhances the secretion of mPL-2 [4]. *Slco2a1* KO maintains the corpus luteum by mPL-2 from the sponge layer of the placenta and causes delayed labor [4].

#### 4.4.2. *Sushi Ichi Retrotransposon Homolog 7/Leucine Zipper, Downregulated Cancer 1* Knockout Mice: *Sirh7/Ldoc1* KO

*Sushi ichi retrotransposon homolog 7/leucine zipper* (*Sirh7/Ldoc1*) is a member of the long terminal repeat (LTR) retrotransposon-like gene family [15]. *Sirh7/Ldoc1* is an X-linked gene that is unique to eutherians; therefore, it is highly conserved in both mice and humans. *Sirh7/Ldoc1* KO showed a delayed delivery of approximately 1–4 days in 49% of mice due to residual P₄ at 18.5 dpc [15]. RU486 and ovariectomy improved prolonged delivery, suggesting that P₄ was the cause of prolonged delivery [15]. The mice had elevated levels of mPL1 as well as placental P₄ production and exhibited a delayed switch from mPL1 to mPL2 during mid-pregnancy [15]. Similar to mPL-2, mPL-1 also has a luteotropic function. MPL-1 has been proposed to be one of the causes of delayed delivery of *Sirh7/Ldoc1* KO [15]. The *Sirh7/Ldoc1* KO placenta also had an irregular boundary between the sponge and labyrinth layers.

#### 4.4.3. *Nik-Related Kinase* Knockout Mouse: *Nrk* KO

In 1996, *Nik-related kinase* (*Nrk*) was named by Kanai-Azuma et al. [47]. Nrk is another X-linked gene that is highly conserved in mice and humans, and its KO mice exhibit three phenotypes: overgrowth of the spongy layer of the placenta, high frequency of breast tumors, and prolonged parturition [16].

Wild-type dams with transplanted *Nrk* KO blastocysts showed delayed parturition, which suggests that the *Nrk* KO fetus or placenta is the cause of delayed delivery [16]. The P₄ and estrogen levels of *Nrk* KO mice were about the same as those of the wild-type at 17.5 and 19.5 dpc [16]. The report postulated that an unidentified soluble factor from the spongy layer protracts parturition [16]. However, they could not identify leading candidates. 

### 4.5. Autoimmune Disorder

Toll-like receptors (TLRs) are receptors on surface of cells such as macrophages and dendritic cells, that bind to different ligands. TLRs recognize pathogens and activate innate immunity. There are 1 to 13 types, each of which is a pattern-recognition receptor that binds to different ligands.

The model of preterm delivery of lipopolysaccharide (LPS) into mice at midgestation has been previously used. TLRs induce labor by activating the NF-κβ pathway and increasing CAPs by recognizing inflammatory substances such as LPS [48]. Therefore, TLR is presumed to be a trigger for pathological preterm delivery due to bacterial infection. TLRs bind to non-inflammatory substances such as Sp-A and damage-associated molecular patterns (DAMPs) emitted by stress, such as uterine stretch [49]. TLRs are also involved in the onset of normal labor.

#### 4.5.1. *Toll-Like Receptor 2*-Deficient Mice: *Tlr2* KO

Compared with wild-type mice, *Tlr2* KO mice showed a significant delay in the timing of labor during the first pregnancy, and a decrease in the amniotic fluid macrophages and Cx43 expression in the myometrium [13]. Sp-A/D diffuses from the fetal lung into the amniotic fluid, and binds to TLR2 in amniotic fluid macrophages [13]. Amniotic fluid macrophages secrete IL1b and IL6, which in turn activate the NF-κB pathway in the uterus and induce labor [13].

#### 4.5.2. *Toll-Like Receptor 4*-Deficient Mice: *Tlr4* KO

Since TLR4 binds to LPS and DAMPs, it is associated with both bacterial infection and aseptic preterm delivery [48,49]. The average gestation length of *Tlr4* KO mice was extended by 13 h [17]. The timing of FP, OTR, and Cx43 expression was delayed [34], and P₄ withdrawal did not occur [17]. Administration of a TLR4 antagonist to wild-type mice resulted in delayed delivery [17]. Therefore, TLR4 is presumed to play an important role in gestation.

#### 4.5.3. *Interleukin 6* Null Mutant Mice: *Il6* KO

*Il6* KO mice delivered 24 h after wild-type mice, yet the P₄ level of the mice was equivalent to that of the wild-type [18]. *Il6* KO delayed the expression of FP and Cox2 in the myometrium, while administration of IL6 to *Il6* KO improved delayed delivery. IL6 is involved in the pathway after P₄ withdrawal and induces PG [18].

### 4.6. Uterine Contractile Dysfunction

Uterine contractile dysfunction refers to the lack of intrauterine pressure sufficient to push a fetus out of the uterus. It is caused by a weak contraction force of the uterine muscles and impairment of coordinated contractions. The intrauterine pressure should be measured directly to accurately observe labor.

There are in vitro and in vivo methods for measuring uterine contractile force. The contractile force of the removed uterine muscle can directly be measured in vitro [33]. Pure muscle contraction force can be measured, but the reproducibility of the in vivo environment is poor.

In vivo, there are two methods: surgical implantation of a pressure transducer and transvaginal insertion into the uterus. For surgical implantation, a pressure transducer is surgically implanted in the myometrium after laparotomy under anesthesia [50]. The advantage of this method is that intrauterine pressure can be continuously observed without anesthesia from pregnancy to delivery. However, special techniques are required, and the operation may cause fetal loss [50]. The transducer insertion method, which has also been used in humans, is less invasion of the dams [51]. This method had three disadvantages: limited insertion time, anesthesia was required at the time of the insertion, and the position of the inserted transducer was not identified [51].

#### 4.6.1. *Connexin 43^fl/fl^:SM-CreER^T2^* Mice: *Sm-CreER^T2^* KO

Delayed delivery was observed in 82% of mice [19]. Uterine contractions were not directly measured. The author reported that the cause of the abnormal delivery was insufficient contraction due to the loss of coordinated contraction of the uterine muscle [19].

#### 4.6.2. *Kcnn3^tm1Jpad^/Kcnn^3tm1jpad^*: *Sk3^T/T^* Mice

The small-conductance Ca^2+^-activated potassium channel isofrom-3 (SK3 channel) is a selective potassium channel. *SK3^T/T^* mice overexpress SK3 channels [20]. The contractile force in these mice was attenuated in vivo [20]. Although the mechanism is unclear, the investigation showed that mice also experience abnormal labor due to weak uterine contractions [20]. It has been reported that the SK3 channel is involved in delivery even in humans [52]. This research is expected to be applied to the treatment of preterm births.

### 4.7. Poor Cervical Ripening

Almost 90% of the cervix is composed of an extracellular matrix, such as collagen and glycosaminoglycans. Cervical ripening is the decomposition of collagen and glycosaminoglycans during late pregnancy. In humans, cervical ripening is evaluated using pelvic examination. In mice, cervical ripening is often evaluated using pathological and biomechanical examinations. There are three pathological examinations: Masson trichrome staining (evaluation of collagen degradation), immunostaining (evaluation of neutrophils and monocytes), and Mayer’s mucicarmine (evaluation of epithelial mucus volume) [22,23,24]. The pressure generated can be determined by mechanically dilating the removed cervix [22,24].

#### 4.7.1. *5α-Reductase Type 1* Knockout Mice: *5αR1* KO

5α-reductase (5αR1) is an enzyme involved in the production of dihydrotestosterone and the decomposition of testosterone. Thirty-three percent of *5αR1* KO mice did not start delivery [23], while 25% of mothers had died [23]. These mice showed a decrease in blood P₄, but administration of RU486 improved delayed labor [23]. Uterine contractions of *5αR1* KO mice were measured by surgical transducer implantation and were not different from those of wild-type mice [22]. The elasticity of the cervix decreased and pathologically, the cervix was poorly mature [22]. It has been speculated that the amount of P₄ in the cervix is increased, resulting in poor cervical ripening and delayed delivery [22].

#### 4.7.2. *TgN(hApoB) 1102SY* Line: *Tg/Tg* Mouse

The *Tg/Tg* mouse is a strain that occurs by the insertion, by chance, of the human apolipoprotein B gene [24]. Mice exhibited delayed labor when the mother was a *Tg/Tg* mouse, regardless of the fetal genotype [24]. P₄ withdrawal was prolonged by one day compared to wild-type mice; administration of a P₄ receptor antagonist (ZK98299) and ovariectomy did not improve prolonged delivery [24]. Uterine contractions were similar to those observed in wild-type mice, the cervix was not elastic, and pathological evaluation showed poor cervical ripening [24].

#### 4.7.3. *Anthrax Toxin Receptor 2* Knockout Mice: *Antxr2* KO

Antxr2 has been postulated to be involved in the proliferation and development of the vascular endothelium, but its function is unknown. *Antxr2* KO mice exhibited a sharp drop in P₄ but did not deliver [21]. The cervix was immature with no collagen degradation, while the myometrium had lost cricothyroid and longitudinal muscle cells [21]. The cause of delayed labor in the mice was poor cervical ripening and uterine contraction failure.

### 4.8. Delayed Implantation

A common method of observing the implantation time is to inject a pigment such as Chicago blue into the tail vein 5 and 6 days after confirmation of the vaginal plug and direct observation of the implantation site [25]. If implantation is delayed, the number of fetuses changes. Delayed implantation often causes a reduction in litter size, fetal dysgenesis, and abnormal spacing.

#### 4.8.1. *Cytosolic Phosphatase A2* Mutant Mouse: *Pla2g4a* KO

Cytosolic phosphatase A2 (Pla2g4a) is an enzyme that produces PGs. *Pla2g4a* KO mice showed prolonged delivery, decreased fetal population, and increased stillbirth rate [26]. The number of implantations in mice increased at 6 dpc compared to that at 5 dpc [25]. The experiment suggested that the implantation period of the mice was delayed [25]. Administration of PGI₂ and PGE₂ to mice improved delayed implantation and prolonged delivery [27].

#### 4.8.2. *Lysophosphatidic Acid Receptor 3*-Deficient Mouse: *LPA3* KO

LPA is a lipid that has a signaling effect and is present in the ovaries, testes, and uterine muscles. There are six types of LPA receptors that have various functions in the reproductive process. Mating between *LPA3* KOs resulted in reduced litter size, delayed implantation, abnormal spacing, and prolonged delivery. In these mice, Cox2, PGE₂, and PGI₂ are decreased in the uterus [28]. Administration of PGE₂ + PGI₂ to mice improved delayed implantation and prolonged parturition; similar to *Pla2g4a* KO, delayed implantation of *LPA3* KO is caused by a decrease in PGs [28]. 

In cattle, LPA promotes the secretion of P₄ and PGE₂ and maintains pregnancy [53]. Therefore, LPA3 may be directly involved in prolonged parturition. It has been reported that in humans, LPA is associated with the maintenance of pregnancy and preterm birth [54]. Therefore, future studies on LPA are required.

### 4.9. Unknown Cause

There are some mice for which the cause is unknown, such as when a delay in labor occurs by chance.

#### 4.9.1. *Erk1 and Erk2 Conditional* Knockout Mouse: *Erk1/2* cKO

Extracellular signal-regulated kinase (ERK) is an intermediate of the MAPK pathway. The MAPK pathway is involved in cell differentiation and cell division. *Erk1* KO was reproductive with no apparent abnormalities, but *Erk2* KO was fatal [55]. *Erk1/2* cKO mice showed abnormal placental development, delayed delivery on average of approximately 2.3 days, and death of offspring [29]. This may have been due to the abnormal placental development [29]. However, the cause of this phenomenon is unknown.

#### 4.9.2. *Leptin* KO Mouse: *Leptin* KO

Leptin is a hormone secreted by fat cells and the placenta and is related to the absorption of nutrients. *Leptin* KO mice were infertile because of an abnormality in the hypothalamic–pituitary system [56]. One study recovered pregnancy and childbirth by administering leptin [30]. *Leptin* KO mice that received leptin throughout gestation increased food intake and became obese compared to the mice that received leptin after that point [30]. The mice showed delayed parturition for approximately 1–3 days; the authors claimed that the cause may have been abnormal obesity [30]. However, to date, no detailed studies have been conducted.

#### 4.9.3. *MT-mER* Mouse

In 1994, Davis et al. published a study in which the estrogen receptor (ER) was forcibly expressed throughout the mouse body. These mice expressed ER in cells expressing the metallothionein-1 promoter. *MT-mER* mice showed delayed delivery for 4 days or more [31]. The *MT-mER* mouse strongly expressed ER in the gonads, but its expression in other tissues was weak. The authors concluded that the cause may have been the difference in the time and place of the ER [31]. However, there is no further description of this cause.

#### 4.9.4. *Activin/Inhibin βB Subunit-Deficient* Mouse: *Activin/Inhibin βB Subunit* dKO

Activin and inhibin are peptides produced in the gonads, pituitary gland, and placenta. The β subunit of activin is common to activin. Previously, activin was presumed to be a substance that induces the mesoderm. However, the *activin/inhibitor βB subunit* dKO showed morphological abnormalities only in the eyelids [32]. The mice showed mildly delayed labor of approximately half a day to a day, while in rats, the administration of activin to the pituitary gland increases blood oxytocin concentrations. Therefore, the authors concluded that a decrease in oxytocin may have been involved [32].

## 5. How to Identify the Cause of Delayed Labor in Mice 

If genetically modified mice exhibit delayed labor, we recommend investigating the items presented in Table 2. This list can help to identify the cause of delayed labor more accurately and efficiently.

### 5.1. Measurement of the Landing Time

Delayed implantation often results in fetal stunting and reduced litter size. Since delayed-labor mice often miscarry or absorb their fetus in utero, measurement of landing time is useful for measuring the true litter size. Some studies have reported that small litter sizes tend to delay delivery in some mouse strains [57].

### 5.2. Investigation of the Following Factors: Dam, Fetus, and Placenta

If the dam is abnormal, mice will exhibit delayed delivery, regardless of the sire genotype. If the placenta or fetus is abnormal, wild-type female transplanting KO embryos, such as *Nrk* KO, will exhibit delayed delivery (16). Investigating the expression of the target gene in the fetus and placenta is also useful to determine whether the cause is the fetus or placenta. Placental dysplasia often occurs if the placenta is the cause. In addition, the method using the above-mentioned LV vector and tetraploid chimera is also useful.

### 5.3. Presence or Absence of P₄ Withdrawal during Late Pregnancy

Abnormal luteolysis is the most common cause of delayed labor; examining P₄ at 16.5 dpc and 18.5 dpc is especially useful. Luteolysis can also be examined by observing the ovaries pathologically.

### 5.4. Evaluation of Uterine Contractions and Cervical Ripening

It is also important to assess uterine contractions and cervical ripening in late pregnancy to directly assess the status of labor. In delayed delivery, there must be an abnormality in either, or both. 

### 5.5. Comparison of Wild-Type Mice

The gestation period of mice varies depending on the strain [58]. However, the variability within the strain is small [58]. Comparison with wild-type mice is necessary to accurately prove the delayed labor. In some mice strains, fetal count and gestational duration are inversely correlated [57,58]. Extension of gestation due to a decrease in the number of fetuses should also be considered.

## 6. Conclusions and Future Perspectives

The etiology of delayed parturition is overexpression of the pregnancy-maintenance mechanism or deficiency of the labor-induction mechanism. Overexpression of the pregnancy-maintenance system is attributed to the excessive secretion of humoral factors, such as P₄ and PL, which have a pregnancy maintenance effect [4,33]. Suppression of the labor-induction mechanism is caused due to uterine contraction disorder and/or cervical ripening [20,21,24,59]. Understanding both mechanisms is important for elucidating the mechanisms of pregnancy and delivery.

The biggest difference between rodents and humans in pregnancy and labor is the trigger of labor. In rodents, P₄ withdrawal triggers labor. In contrast, in humans, P₄ withdrawal does not occur in late pregnancy, suggesting that the trigger for labor may be a PR subtype change [39,42]. P₄ administration to high-risk patients with preterm birth reduces preterm birth rates [60]. The human corpus luteum disappears at approximately 20 weeks of gestation, and P₄ is produced in the placenta. Luteolysis in humans does not appear to be involved in delayed labor. Mice with delayed parturition due to abnormal luteolysis may be useful to understand human early pregnancy maintenance.

Human PL, which is only one type of PL, does not affect pregnancy [61]. There are several types of mouse PLs, and mPL-1 and mPL-2 maintain the corpus luteum and pregnancy [4,15]. The functions of other PLs are unknown, and further research is required.

However, few studies have evaluated the effects of cervical ripening and uterine contractions in mice. In humans, cervical ripening is evaluated on the basis of the degree of cervical dilatation. Uterine contractions are evaluated using cardiotocography. Similarly, examining these two factors in mice will help to clarify the cause of labor disorders.

The length of the gestation period is controlled not only by the maternal signals but also by fetal and placental signals [4,13,14,15]. The amount of mPL changes with placental and fetal growth [4,15]. The concentration of Sp-A in amniotic fluid increases with fetal maturation and promotes labor [13]. In humans, corticotropin-releasing hormone secreted from the fetal hypothalamus promotes labor induction [42]. The mother controls these two systems by sensing fetal development.

Studies on preterm birth in humans, which has a greater effect on infants, are more abundant than those on delayed labor. However, findings from studies on delayed delivery can also be applied to the prevention and treatment of preterm births. For example, *Fp* KO and Cox1 KO has been used to study cox-selective inhibitors [35]. P4-related delayed delivery mice such as *Klf 9* KO and *Cyp11a1* Tg provide clues to the potential of P4 as a treatment for preterm birth. P₄ has been used in the treatment of imminent preterm births in humans [60]. Delayed delivery mice due to autoimmune disorders may be associated with surrogate mothers and pregnancy due to uterine transplantation. In humans, delayed labor has a family accumulation and is likely to involve a genetic background. In the future, if the genetic background of women who are prone to delayed delivery is clarified, it may be possible to find an association with the gene related to these mice.

Elucidation of the causes of delayed parturition in humans and animals is essential to understand the mechanisms underlying pregnancy and delivery. Summarizing the causes and examination methods of delayed delivery in mice in this review will help to focus on future research and provide new insights into novel treatment strategies.

## Figures and Tables

**Figure 1 jdb-10-00020-f001:**
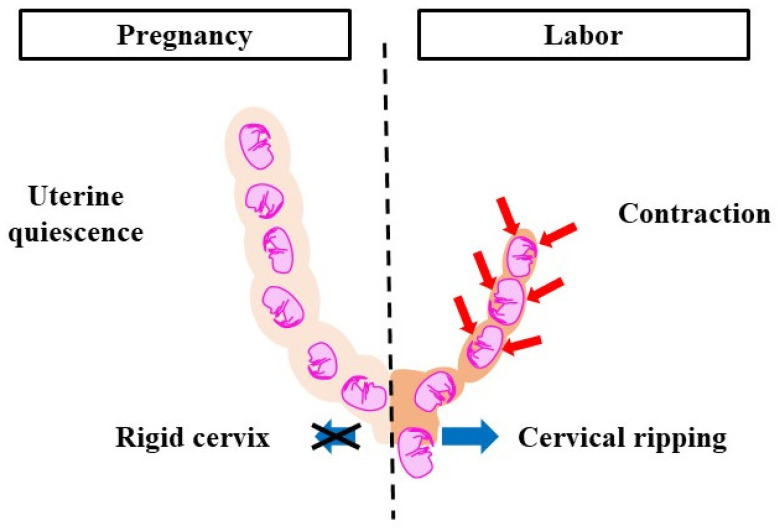
Pregnancy-maintenance mechanism and labor-induction mechanism. During pregnancy, uterine contractions do not occur and the cervical canal is immature. During labor, uterine contractions are activated and cervical ripening is promoted. To deliver at term, the pregnancy-maintenance mechanism and labor-induction mechanism need to work at the right time. Red arrows: contraction, blue arrows: cervical dilation.

**Figure 2 jdb-10-00020-f002:**
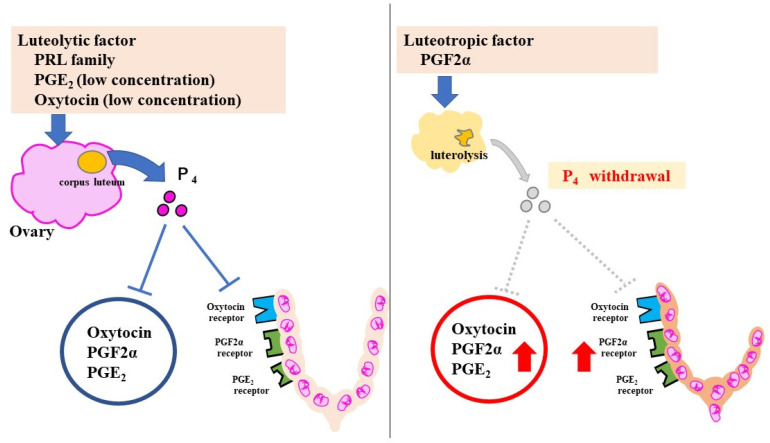
Pregnancy-maintenance and parturition mechanisms in mice. The luteolytic factor maintains the corpus luteum during pregnancy. P₄ suppresses CAPS. Luteolysis at 18.5 dpc causes a sharp decrease in P₄. With an increase in CAPS, uterine contractions and cervical ripening progress rapidly. The red Arrows indicate increased effect.

**Figure 3 jdb-10-00020-f003:**
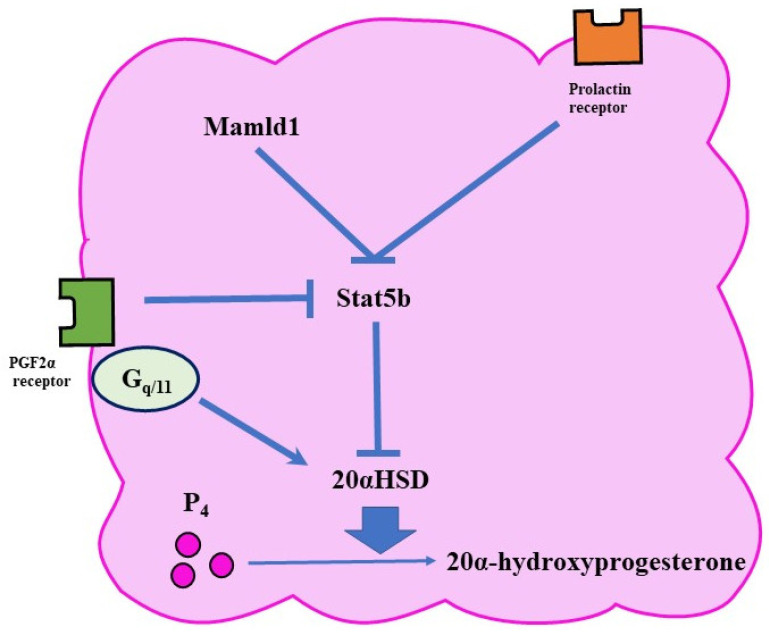
Functional luteolysis. Functional luteolysis is caused by the inactivation of P₄ by 20αHSD. There are various pathways for regulating the expression of 20αHSD, such as the FP and prolactin-receptor pathways.

**Figure 4 jdb-10-00020-f004:**
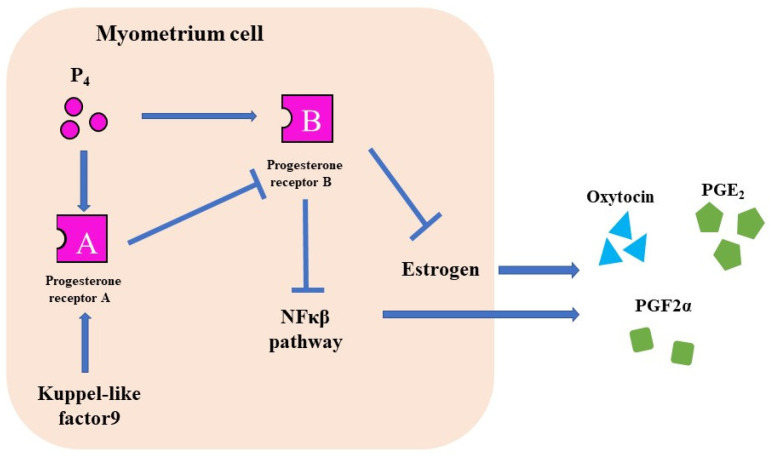
The function of P₄ receptors. There are two types of PRs. PR-B suppresses CAPS and maintains pregnancy. PR-A induces labor induction by suppressing the PR-B.

**Table 1 jdb-10-00020-t001:** Classification of genetically modified mice that exhibit delayed delivery (official name: abbreviation). We searched the literature published between 1970 and 2020, and classified the 26 mice according to the cause of delayed labor.

1. Failure of luteolysis *Prostaglandin F* Receptor knockout mouse: *Fp* KO [5,6] *Cyclooxygenase-1*-deficient mouse: *Cox1* KO [7] *20α hydroxysteroid dehydrogenase*-deficient mice: *20α Hsd* KO [8] *Gα_q_^f/f^;Gα_11_^-/-^;Cre^+^* mice: *Gq/11* cKO [9] *Mastermind-like domain-containing 1* knockout mouse: *Mamld1* KO [10]
2. Abnormalities in progesterone metabolism and receptors *Kruppel-like factor 9* knockout mouse: *Klf 9* KO [11] *Cytochrome p450 family 11a1*-overexpression mouse: *Cyp11a1* Tg [12]
3. Fetal factors *Surfactant protein A and D* double-deficient mice: *Sp A/D* dKO [13] *Steroid receptor coactivator 1 and 2* double-deficient mice: *Src 1/2* dKO [14]
4. Placental factors *Solute carrier organic anion transporter family member 2A1* knockout mouse: *Slco2a1* KO [4] *Sushi-ichi retrotransposon homolog 7/Leucine zipper, downregulated cancer 1* knockout mouse: *Sirh7/Ldoc1* KO [15] *Nik-related kinase* knockout mouse: *Nrk* KO [16]
5. Autoimmune disorder *Toll-like receptor 2*-deficient mice: *Tlr* 2 KO [13] *Toll-like receptor 4*-deficient mice: *Tlr* 4 KO [17] *Interleukin 6* null mutant mice: *Il6* KO [18]
6. Uterine contractile dysfunction *Connexin 43^fl/fl^:SM-CreER^T2^* mice: *Sm-CreER^T2^* KO [19] *Kcnn3^tm1Jpad/^Kcnn^3tm1jpad^: Sk3^T/T^* mice [20] *Anthrax toxin receptor 2* knockout mice: *Antxr 2* KO [21]
7. Poor cervical ripening *5α reductase type 1* knockout mice: *5αR1* KO [22,23] *TgN(hApoB)1102SY* line: *Tg/Tg* mouse [24] *Anthrax toxin receptor 2* knockout mice: *Antxr 2* KO [21]
8. Delayed implantation *Cytosolic phosphatase* *A2* mutant mouse: *Pla2g4a* KO [25,26,27] *Lysophosphatidic acid receptor 3*-deficient mouse: *LPA3* KO [28]
9. Unknown cause *Erk 1 and Erk 2* conditional knockout mouse: *Erk1/2* cKO [29] *Leptin* KO mouse: *Leptin* KO [30] *Mt mER* mouse [31] *Activin/inhibin βB subunit*-deficient mouse: *Activin/inhibin βB subunit* dKO [32]

**Table 2 jdb-10-00020-t002:** List for finding the cause of delayed delivery. If genetically modified mice exhibit delayed labor, we recommend investigating the items.

Before pregnancy	Transplanting KO embryos into wild-type mice Tetraploid chimera Administration of LV
5.5–6.5 dpc	Delay implantation
17.5–19.5 dpc	Measurement of P_4_ Evaluation of cervical ripening (biometric test, pathological test) Evaluation of uterine contraction (in vivo and vitro)
19.5 dpc-	HE staining and/or tunnel staining of corpus luteum Administration of RU486 Ovariectomy
After labor	Investigation of the maternal and fetal genotype with delayed labor Screening of placenta and fetus

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
