# Peer review of "A Review of Delayed Delivery Models and the Analysis Method in Mice"

_jdb, 2022, doi:10.3390/jdb10020020_

Round 1
Reviewer 1 Report
Overall this is a very good review of delayed implantation.
A few areas would increase the significance of this review
1) Explain the differences in the time of delivery of different mouse strains (C57Bl6 vs ICR) etc
2) Do different strains of rats deliver at different delivery times ?
3) Add a section on the differences between rodents and humans - this is a very important distinction and should speak to the relevance of animal models
4) Add a section on which of the animal models discussed in this review have actual clinical relevance to delayed delivery in humans. This is an important section and should be a strength of the review
Reviewer 2 Report
The authors present a overview of the factors involved in delayed delivery. The reviewer found this manuscript well written, easy to follow and comprehensive.
The objective of the paper to review mice model of delayed delivery models is well defined and the authors described multiple relevant mouse models and described with enough details the known mechanisms associated with each models.
To me this manuscript is original and could be a seminal paper for researcher wanting to use delayed pregnancy models.
Although the other reviewer found that human relevance is poorly described, I think it is outside the scope of this review (description of mouse model and pathway analysis).
Thus, I would accept the publication of this manuscript.
Minor comments:
Line 98 : Is the role of gap junctions samilar or different than Cx?
Line 46 was-> is
